# Endosomal sorting protein SNX4 limits synaptic vesicle docking and release

**Josse Poppinga[1], Nolan J Barrett[1], L Niels Cornelisse[1,2], Matthijs Verhage[1,2], Jan RT van Weering[1,2]\***

[1]Department of Functional Genomics, Center for Neurogenomics and Cognitive Research (CNCR), VU University, Amsterdam, Netherlands; [2]Department of Human Genetics, CNCR, Amsterdam UMC, Amsterdam, Netherlands

## eLife Assessment

This **important** study presents a series of results aimed at uncovering the involvement of the endosomal sorting protein SNX4 in neurotransmitter release. While the evidence supporting the conclusions is **solid**, the molecular mechanisms remain unclear. This paper will be of interest to cell biologists and neurobiologists.

**\*For correspondence:**
jan.van.weering@vu.nl

**Competing interest:** The authors declare that no competing interests exist.

**Abstract** Sorting nexin 4 (SNX4) is an evolutionary conserved organizer of membrane recycling. In neurons, SNX4 accumulates in synapses, but how SNX4 affects synapse function remains unknown. We generated a conditional SNX4 knock-out mouse model and report that SNX4 cKO synapses show enhanced neurotransmission during train stimulation, while the first evoked EPSC was normal. SNX4 depletion did not affect vesicle recycling, basic autophagic flux, or the levels and localization of SNARE-protein VAMP2/synaptobrevin-2. However, SNX4 depletion affected synapse ultrastructure: an increase in docked synaptic vesicles at the active zone, while the overall vesicle number was normal, and a decreased active zone length. These effects together lead to a substantially increased density of docked vesicles per release site. In conclusion, SNX4 is a negative regulator of synaptic vesicle docking and release. These findings suggest a role for SNX4 in synaptic vesicle recruitment at the active zone.

## Introduction

Endosome sorting plays a critical role in the recycling and degradation of proteins in various eukaryotic cell types, including neurons (**Cullen and Steinberg, 2018**; **Huotari and Helenius, 2011**; **Sardana and Emr, 2021**; **Vietri et al., 2020**). Synapses, in particular, rely on rapid recycling mechanisms to replenish synaptic vesicles (SVs) and facilitate neurotransmitter release (**Kononenko and Haucke, 2015**; **Rizzoli, 2014**; **Südhof, 2000**). The extent to which endosome sorting is involved in these processes has been a subject of ongoing debate since the early studies on SV recycling (**Ceccarelli et al., 1973**; **Heuser and Reese, 1973**; **Kononenko and Haucke, 2015**; **Watanabe and Boucrot, 2017**). While it has been demonstrated that depleting endosomal proteins affect neurotransmitter release (**Chen et al., 2017**; **Tagliatti et al., 2016**; **Uytterhoeven et al., 2011**), the precise role of these sorting proteins on SV maintenance and fusogenity remains not well understood.

Sorting nexins (SNXs) are evolutionary conserved organizers of endosome sorting that are recruited to endosome membrane domains by a phox homology (PX) domain (**Carlton et al., 2005**). SNX4 belongs to a subgroup of SNXs containing a C-terminal Bin-Amphiphysin-Rvs (BAR) domain that recognizes and remodels curved endosome membranes (**Cullen and Steinberg, 2018**; **Teasdale et al., 2001**; **van Weering et al., 2012a**). While several SNX-BARs are associated with the retromer-related

Endosomal SNX-BAR Sorting Complex for Promoting Exit (ESCPE)-1 (*Simonetti et al., 2023*), SNX4 forms a distinct sorting complex with SNX7, SNX30 (*Antón et al., 2020*; *van Weering et al., 2012b*), SNX5 (*Zhou et al., 2022*), or SNX32 (*Sugatha et al., 2023*) for endosome-to-cell surface recycling pathways and autophagosome traffic (*Traer et al., 2007*; *van Weering et al., 2012a*; *Zhou et al., 2022*). SNX4 emerges as a central hub in this endosome recycling pathway as the expression of SNX4 is required for the stable SNX7 and SNX30 protein levels in cells (*Antón et al., 2020*). SNX4 is expressed in all cell types but to what extent this conserved principle also operates in highly polarized mammalian cells, such as neurons, remains unclear. In post-mortem brain material of Alzheimer's disease patients, altered protein levels of SNX4 have been observed (*Kim et al., 2017*). SNX4 is implicated in the recycling of the amyloid precursor protein cleaving enzyme (BACE-1), sorting it away from the endolysosomal pathway (*Kim et al., 2017*; *Toh et al., 2018*). Consequently, dysregulation of SNX4 may lead to the mislocalization of BACE-1 to late endosomal compartments, potentially contributing to altered amyloid-β-production. In yeast, Snx4 is essential for the endosome recycling of Snc1p, a homologue of the SNARE-protein VAMP2/synaptobrevin-2 (VAMP2/syb2) that is essential for synaptic transmission (*Hettema et al., 2003*; *Ma et al., 2017*; *Schoch et al., 2001*). In neurons, SNX4 specifically accumulates in synapses, and its knockdown dysregulates protein pathways involved in neurotransmission (*Vazquez-Sanchez et al., 2020*). While its synaptic subcellular localization has been shown, the functional role of SNX4-dependent endosome sorting in synapses remains unknown.

In this study, we characterize the role of endosomal sorting protein SNX4 in synapse function using a novel conditional SNX4 knock-out mouse model. We show that SNX4 limits SV docking and release. SNX4 depletion enhances neurotransmission specifically during train stimulation, while the first evoked response is not significantly affected. SNX4 depletion does not impact vesicle recycling or the levels and localization of the SNARE-protein VAMP2/syb-2. The overall number of SVs is unaffected in SNX4 cKO neurons, but these synapses show an increased number of docked SVs at the active zone of the presynapse. The active zone length is decreased, thereby further increasing the density of SVs at the release site of SNX4 cKO synapses. Our data collectively shows that the endosomal sorting protein SNX4 is a negative regulator of SV docking and release.

## Results

### Neuronal SNX4 protein levels are depleted in 21 days in conditional SNX4 knock-out mouse primary cultures

We generated a conditional knock-out mouse model for SNX4 (*Figure 1a* and Materials and methods section) to circumvent previously reported off-target effects using shRNA approaches (*Vazquez-Sanchez et al., 2020*). In neuronal cultures, SNX4 has a half lifetime of 10.6 days (*Dörrbaum et al., 2018*). To test SNX4 optimal knock-out efficiency, cKO neurons were fixed after 21 days in vitro (DIV21) and Cre expression was initiated at different time points (DIV7, DIV3, and DIV0) (*Figure 1b*). SNX4 expression was detected by western blot which showed a reduction of 67% after 14 days of Cre, 83% after 18 days of Cre, and 90% after 21 days of Cre compared to control neurons infected with delta-Cre (*Figure 1c and d*). Immunohistochemistry of SNX4 in primary hippocampal neurons showed a 64% reduction in SNX4 in MAP2-neurites after 21 days of expressing Cre (*Figure 1e and f*). Morphological parameters such as dendrite length and dendrite branching, based on MAP2 staining, and synapse number, based on VAMP2 puncta, remained unaffected in SNX4 cKO neurons (*Figure 1f–i* and Figure 4a). These data show a successful depletion of 90% of endosomal sorting protein SNX4 using a novel conditional knock-out mouse model for SNX4.

### SNX4 depletion enhances neurotransmission during train stimulation

To investigate the functional role of SNX4 in synapses, we studied the effect of SNX4 depletion on neurotransmission by electrophysiological recordings in single hippocampal neurons cultured on glia micro-islands (autapse) at DIV21 (*Figure 2a*). Both the amplitude and charge of the first evoked response showed a non-significant trend toward an increase in SNX4 cKO neurons (KO) compared to neurons infected with delta-Cre (control) (*Figure 2b, c, and d*). The rescue condition where SNX4 was re-expressed at DIV14 (KO+SNX4) showed a decreased amplitude compared to KO but not to control. A similar trend was observed for total charge (*Figure 2b and d*). Paired pulse ratios were not affected in SNX4 cKO neurons (*Figure 2e*). We next investigated the effect of SNX4 depletion

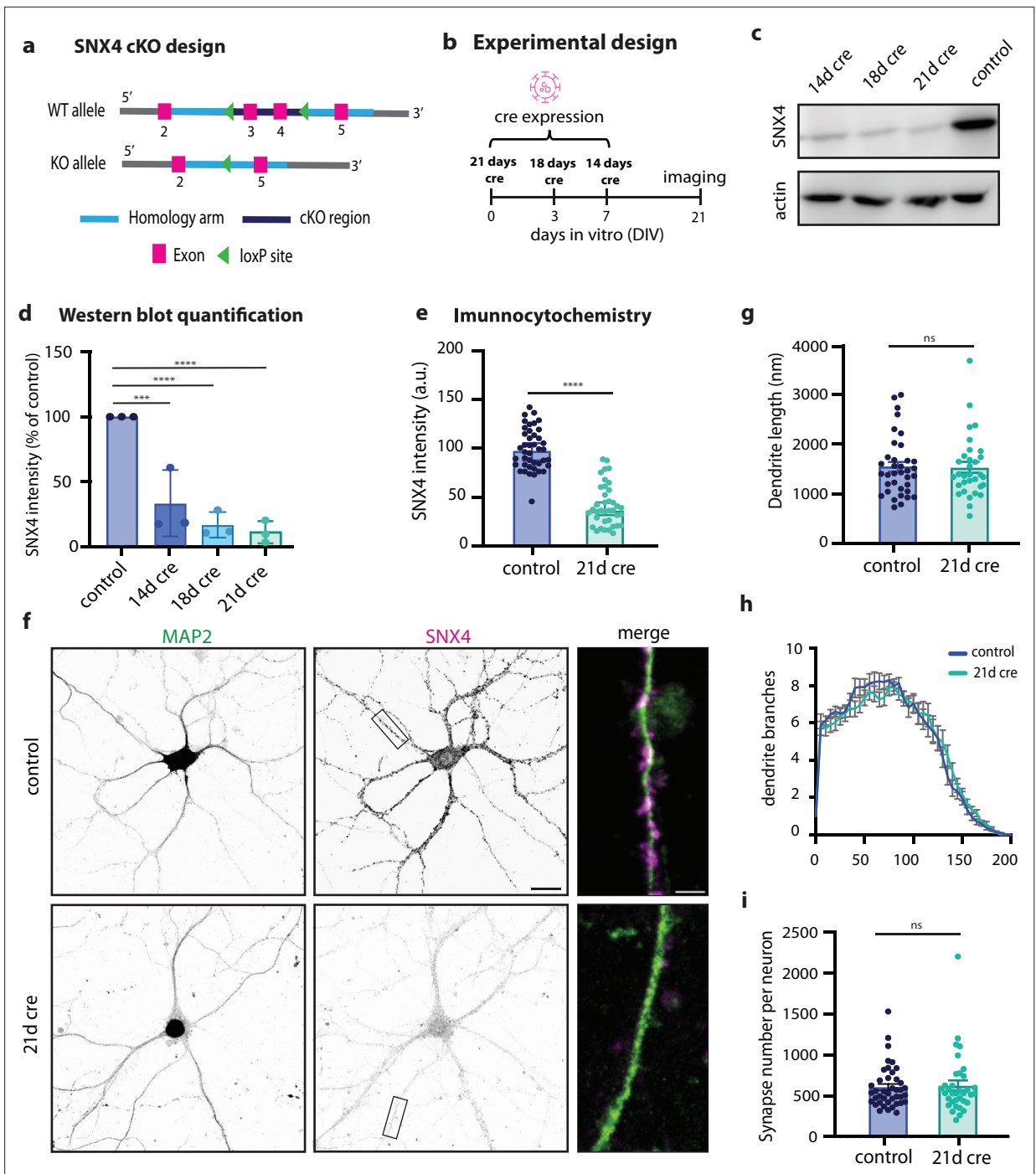

**Figure 1.** SNX4 is knocked out in primary hippocampal neurons using a novel conditional knock-out mouse line. (**a**) Schematic overview of conditional knock-out design of SNX4. (**b**) Schematic overview of experimental design using Cre intervention at different DIVs. (**c**) Typical example of western blot of SNX4 and actin at 14 days, 18 days, and 21 days of Cre and control (delta-Cre) expression. (**d**) Quantification of western blot SNX4 protein levels for control (dcre), 14 days of Cre, 18 days of Cre, and 21 days of Cre, normalized to control. (**e**) Quantification of immunocytochemistry SNX4 intensity normalized to control. (**f**) Typical example of primary hippocampal neurons stained for SNX4 (magenta) and MAP2 (green) for control and 21 days of Cre. Scale bar = 20 μm. Zoom-in scale bar = 3 μm. (**g**) Dendrite length for control and KO hippocampal neurons. (**h**) Sholl analysis for number of dendrite branches. (**i**) Number of synapses per neuron (MAP2 mask) based on VAMP2 puncta (shown in **Figure 4**). ****p=<0.0001, ***p=0.0003, **p=0.0005, *p=0.0019. Ordinary one-way ANOVA with Tukey's multiple comparisons test for (**d**) with N=3 animals. Mann-Whitney test for (**e**, **g**, and **h**) with n=32–42 neurons per group and N=6 animals. Data points represent individual western blots/neurons; bar graph represents mean with SEM.

The online version of this article includes the following source data for figure 1:

*Figure 1 continued on next page*

*Figure 1 continued*

**Source data 1.** Original files for western blot analysis displayed in *Figure 1c*.

**Source data 2.** EPS file of the full western blots displayed in *Figure 1c*.

on synaptic transmission during sustained activity using a series of train stimulations at 5, 10, and 40 Hz (results in *Supplementary file 1*, *Figure 2—figure supplement 2*). During a train of 100 action potentials (APs) at 40 Hz, the charge in SNX4 cKO neurons was highly increased (*Figure 2f and h*). The average charge between pulse 90 and 100 was increased by 110% in SNX4 cKO synapses compared to control (*Figure 2g*). SNX4 overexpression rescued this increase in charge to wild-type (WT) levels (*Figure 2f, g, and h*). The size and refilling of the readily releasable vesicle pool (RRP) was estimated using back extrapolation of the cumulative charge (*Figure 2h*; *Schneggenburger et al., 1999*). While we found no significant differences in RRP estimate size in SNX4 cKO synapses compared to control (*Figure 2i*), the slope estimate of the cumulative charge was increased upon SNX4 depletion, which was rescued by SNX4 overexpression (*Figure 2j*). The amplitude and frequency of spontaneous miniature EPSCs (mEPSCs) were not affected upon SNX4 depletion (*Figure 2k–m*). In order to investigate a direct effect of SNX4 overexpression on neurotransmission, we expressed exogenous SNX4 on WT control background. Both the amplitude and charge of the first evoked response showed no effect in control+SNX4 neurons compared to control, and no differences were detected in the response to the 40 Hz stimulation train (*Figure 4—figure supplement 1a–e*), indicating that SNX4 overexpression in itself does not affect the neurotransmission protocols studied in SNX4 cKO experiments. Together these data show enhanced neurotransmission during train stimulation in SNX4 cKO autapses, while single EPSCs, mEPSCs, and the RRP size estimate remain unaffected. This suggests that vesicle recruitment during high-frequency train stimulation is limited by SNX4.

## SV recycling and retrieval are not altered in SNX4 cKO neurons

As the recruitment rate of releasable vesicles during train stimulation was increased in SNX4 cKO neurons, we assessed its role in SV recycling using the SV fusion reporter synaptophysin-superecliptic-pHluorin (SypHy) (*Granseth et al., 2006*). Neurons were stimulated by two trains of 100 APs at 40 Hz. The total pool of SypHy in acidified SVs per synapse was determined by NH4 superfusion (*Figure 3a and b*). SNX4 cKO neurons showed a higher fluorescence intensity upon both stimulation trains (*Figure 3c*), represented as the stimulated fraction (*Figure 3d and e*), and return to baseline fluorescence level within 60 s over a total imaging time course of 160 s (*Figure 3c*). The fluorescence decay time constant tau in the 60 s after electrical stimulation, a measure for SV endocytosis, was not changed after both stimuli in SNX4 cKO neurons (*Figure 3f*). Finally, the responding fluorescent puncta during AP stimulation as a ratio of the total fluorescent puncta during NH4, the percentage of active synapses, showed a trend toward fewer active synapses in SNX4 cKO neurons (*Figure 3g*). Hence, these data show that SNX4 depletion enhances SV release without altering SV recycling and the number of active synapses in hippocampal neurons.

It has, however, been shown that different SV recycling pathways dominate at different temperatures (*Watanabe et al., 2013*). To test if SNX4 affects SV recycling at physiological temperatures, we used a FM4-64 dye loading protocol before electrical field stimulation (*Figure 3h*). Recycling SVs were labeled by a 60 s incubation with FM4-64 in 60 mM K$^+$ at 37°C. Vesicle fusion was triggered by a train of APs at 40 Hz, resulting in a decreased FM4-64 fluorescence intensity (*Figure 3h and i*). SNX4 cKO neurons showed a similar fluorescence intensity decay after electrical stimulation over a total imaging time course of 80 s (*Figure 3j*). The average fluorescence signal in the last 10 s normalized to baseline (F/F0), a measure for the total amount of recycling SVs fusing, was not significantly different between control and SNX4 cKO neurons (*Figure 3k*). FM4-64 uptake, a measure for endocytosis at 37°C, was also not altered in SNX4 cKO neurons (*Figure 3l*). Taken together, these data suggest that SNX4 depletion does not affect SV endocytosis and recycling at both room temperature and physiological temperature.

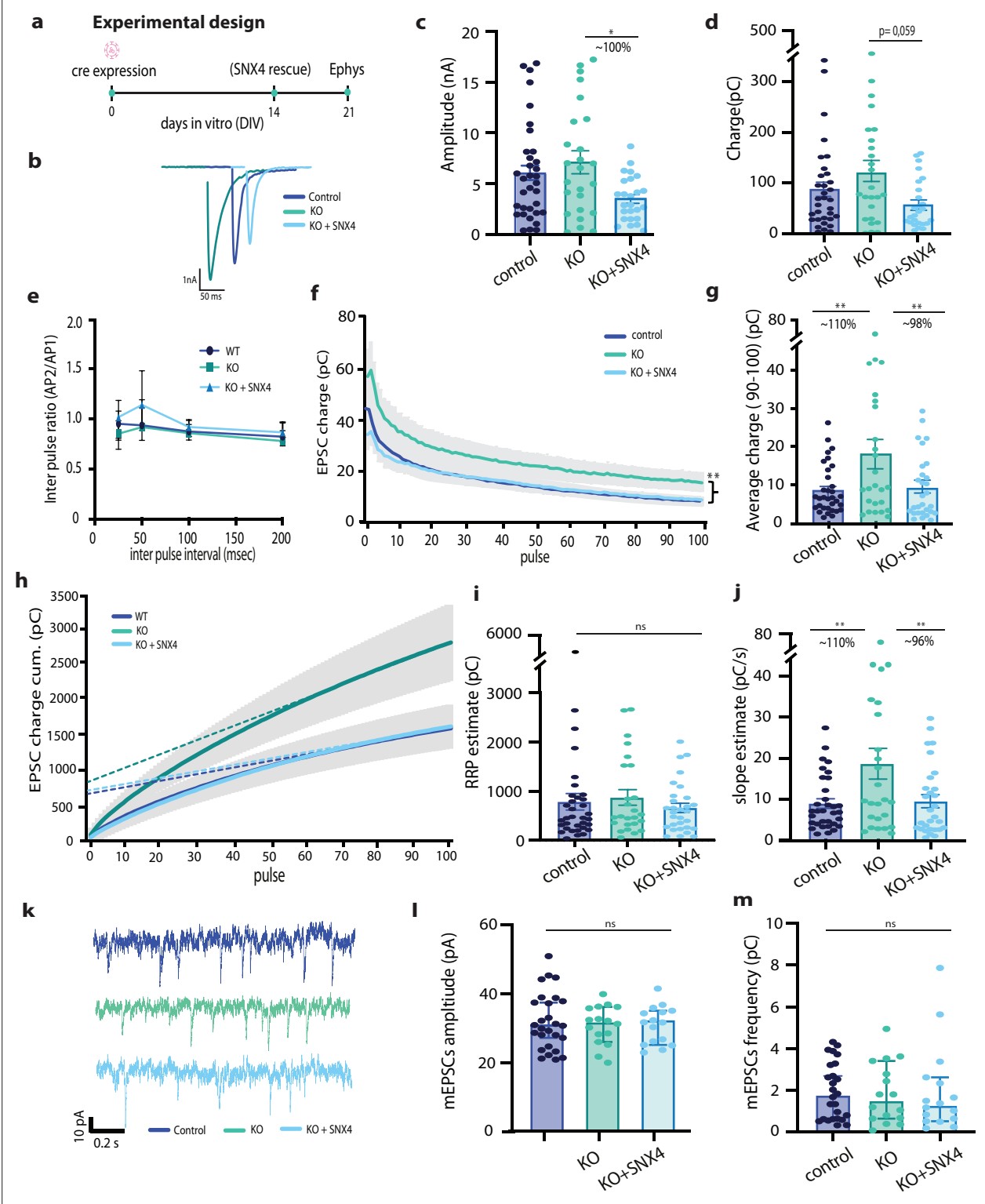

**Figure 2.** SNX4 depletion increases sustained synaptic vesicle release. (**a**) Schematic overview of experimental design using Cre intervention at days in vitro (DIV)0, SNX4 rescue at DIV14, and recording at DIV21. (**b**) Representative traces of the first evoked response for control, KO, and KO+SNX4 upon stimulation. (**c**) Amplitude of first evoked response. (**d**) Charge of first evoked response. (**e**) Paired pulse ratios of amplitude at inter pulse intervals of 25, 50, 100, and 200 ms. (**f**) Mean EPSC charge trace for 100 action potentials (APs) at 40 Hz train. (**g**) Average charge at pulse 90–100. (**h**) Cumulative EPSC charge for 100 APs at 40 Hz train with back extrapolation linear fit to APs 80–100. (**i**) Estimate of RRP based on back extrapolation (**h**). (**j**) Estimate of slope based on back extrapolation. (**k**) Representative traces of miniature EPSC (mEPSC) response for control, KO, and KO+SNX4. (**l**) mEPSC amplitude,

*Figure 2 continued on next page*

*Figure 2 continued*

(**m**) mEPSC frequency. Multilevel ANOVA with n=25–38 neurons per group of N=3–4 animals. \*\*\*p<0.0001, \*\*p<0.001. Data points represent individual neurons; bar graph represents mean with SEM.

The online version of this article includes the following figure supplement(s) for figure 2:

**Figure supplement 1.** EPSC charge for 5 and 10 Hz train stimulation.

**Figure supplement 2.** SNX4 overexpression does not affect first evoked EPSC or neurotransmission during sustained release.

## SNX4 depletion does not affect synaptic VAMP2/syb-2 levels and distribution or autophagy markers in neurons

In yeast, it is reported that SNX4 homologue Snx4p mediates retrieval of Snc1p, yeast's VAMP2/syb-2 homologue (*Hettema et al., 2003*; *Ma et al., 2017*). Missorting of VAMP2 in neurons could affect synaptic transmission. Therefore, we investigated VAMP2/syb-2 as potential cargo of SNX4. To test whether SNX4 mediates VAMP2/syb-2 retrieval in neurons, we investigated the localization, number, and expression levels of VAMP2/syb-2 upon SNX4 knock-out by immunohistochemistry. SNX4 immunostaining showed clear co-localization with VAMP2/syb-2 (*Figure 4a*). In SNX4 cKO neurons, no significant difference was found in VAMP2/syb-2 puncta number per µm dendrite or VAMP2/syb-2 intensity (*Figure 4a, b, and c*). Sholl analysis of average VAMP2/syb-2 intensity over distance from soma also revealed no differences in VAMP2/syb-2 localization upon SNX4 depletion (*Figure 4d*). Western blot analysis revealed no changes in total VAMP2/syb-2 protein levels (*Figure 4e*). These data show that SNX4 is not required for a normal VAMP2/syb-2 distribution and protein level, as observed for the yeast orthologs of both proteins. Other potential cargos of SNX4 that could affect neurotransmission include those involved in the autophagy pathway (*Antón et al., 2020*; *Kuijpers et al., 2021*). We investigated autophagy markers and autophagosomes following SNX4 depletion, using P62 as a basal autophagic receptor marker and bafilomycin to inhibit autolysosome formation, thereby assessing basal autophagic flux. In SNX4 cKO neurons, there was no significant difference in P62 puncta numbers or P62 somatic intensity under basal conditions or after bafilomycin treatment, suggesting that autophagic flux remains normal (*Figure 4—figure supplement 1a–c*). Western blot analysis also showed no changes in total ATG5 protein levels (*Figure 4—figure supplement 1d and e*) and ultrastructural analysis revealed no differences in the total number of autophagosomes (*Figure 4—figure supplement 1f and g*). Collectively, these data indicate that SNX4 depletion does not impact the basal autophagic flux, ATG5 protein levels, or the number of autophagosomes.

## Presynaptic ultrastructure and SV docking are affected in SNX4 cKO synapses

Next, we sought to further investigate synapse organization upon SNX4 depletion using electron microscopy (*Figure 5a*). The number of docked vesicles was measured by direct contact of vesicles with the active zone plasma membrane (*Figure 5a and b*). The average SV size, defined by all clear core vesicles smaller than 50 nm, was reduced by 5%, whereas vesicles larger than 50 nm did not show significant changes upon SNX4 depletion (*Figure 5c and d*). Additionally, the distribution of vesicles, grouped by bin size of 10 nm, showed no notable differences upon SNX4 depletion. (*Figure 5—figure supplement 1*). In SNX4 cKO synapses, the number of docked vesicles was significantly higher than in control (*Figure 5a and f*) while the total number of vesicles was not altered (*Figure 5e*). When normalized to the active zone length, the density of docked vesicles at the active zone length further increases by 50% in SNX4 cKO synapses compared to control (*Figure 5g*). Moreover, we observed a significant decrease in active zone length, PSD length, and vesicle cluster area (*Figure 5g–j*). Together these data show that SNX4 depletion affects the presynaptic ultrastructure, characterized by smaller active zone length and vesicle cluster area, suggesting a decrease in overall synapse size. In these smaller synapses, SNX4 cKO neurons contain more docked SVs leading to a substantially increased density of docked vesicles per release site.

## Discussion

In this study we addressed the role of SNX4-dependent endosome sorting in synapses, using a new conditional SNX4 knock-out mouse model. We show that SNX4 cKO synapses elicit enhanced

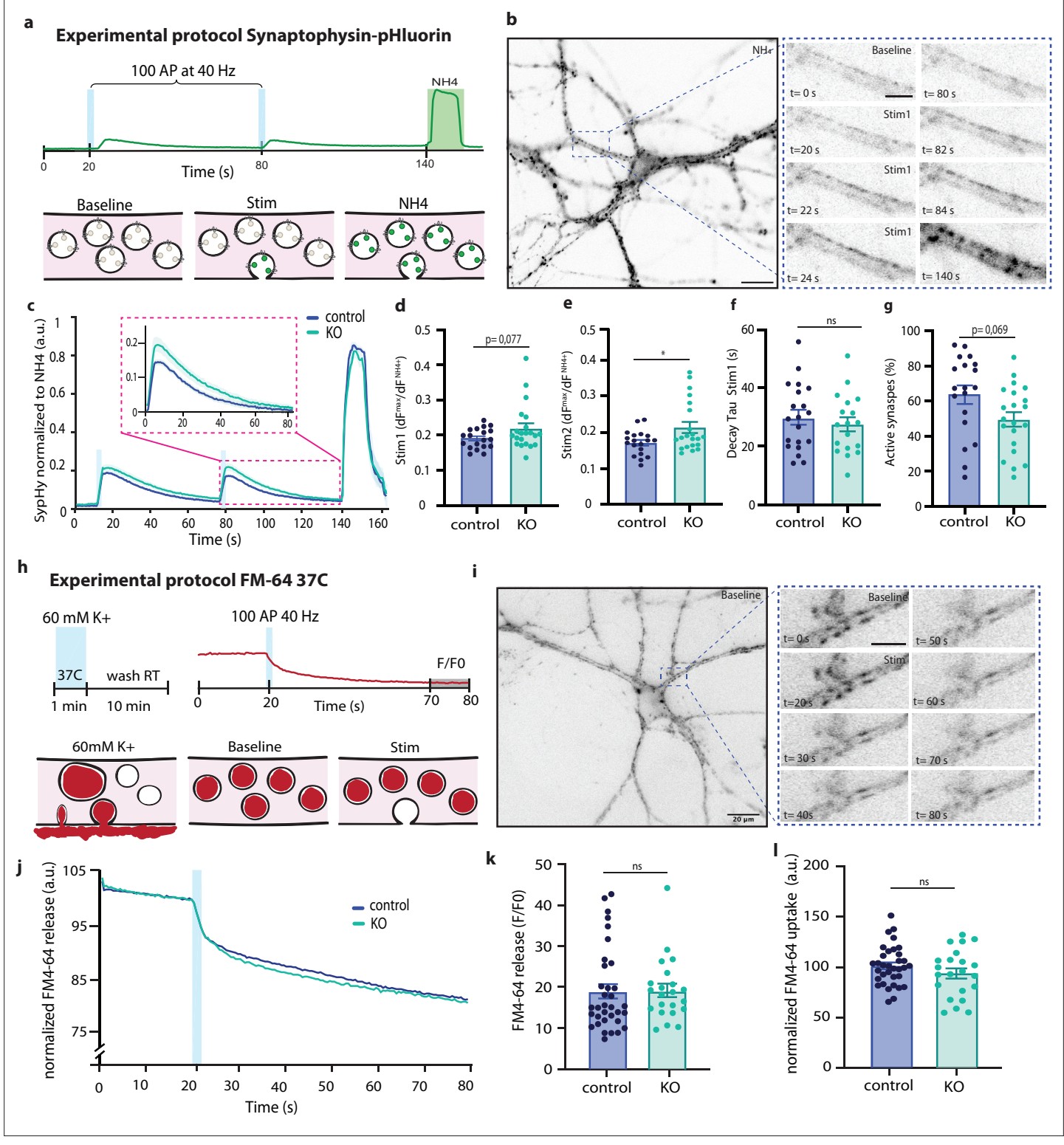

**Figure 3.** SNX4 depletion does not affect synaptic vesicle (SV) recycling at room temperature and physiological temperatures. (**a**) Stimulation paradigm with a single vesicle fluorescence intensity trace. (**b**, Left) Representative image of a synaptophysin-super-ecliptic-pHluorin (SypHy)-infected neuron during Tyrode's NH4 application (scale bar: 20 µm). (Right) Neurite before (baseline) and during (100 action potentials [APs] at 40 Hz) stimulation and during NH4 application (NH4) (scale bar: 10 µm). (**c**) The average fluorescence normalized from baseline to maximum SypHy intensity during Tyrode's NH4 application. (**d**) Maximum response amplitude during the first electrical stimulation normalized to maximum NH4 response plotted as $\Delta F^{max}/F^{NH4}$. (**e**) Maximum response amplitude during the second electrical stimulation normalized to maximum NH4 response plotted as $\Delta F^{max}/F^{NH4}$. (**f**) Fluorescence

*Figure 3 continued on next page*

*Figure 3 continued*

decay time constant tau in the 60 s after electrical stimulation representing SV endocytosis and acidification. (**g**) Percentage active synapses. (**h**) Stimulation paradigm with a single vesicle fluorescence intensity trace. Recycling SVs are labeled by 60 s incubation with FM4-64 in 60 mM K$^+$ at 37°C, followed by a 10 min washout. Electrical field stimulation (100 APs, 40 Hz) triggers SV exocytosis recorded as decreased FM4-64 fluorescence intensity. (**i, Left**) Representative image of FM64-dye endocytosed by a neuron during baseline (scale bar: 20 μm). (Right) Neurite before (baseline) and during (100 APs at 40 Hz) stimulation (scale bar: 10 μm). (**j**) The average FM4-64 fluorescence traces, normalized to control baseline, over time showing initial SV loading, followed by release after electrical field stimulation. (**k**) FM4-64 release upon electrical field stimulation. (**l**) FM4-64 uptake (upon 60 mm K$^+$ incubation). Multilevel ANOVA with N=3 animals and n=21–24 neurons per group for SypHy experiment and N=4 animals with n=28–36 neurons per group for FM64-dye experiment. Bar graph represents mean with SEM. Data points represent individual fields of view.

neurotransmission during train stimulation. SNX4 depletion did not affect vesicle recycling, basal autophagic flux, or the levels and localization of SNARE-protein VAMP2/syb-2. Electron microscopy showed that synapses had an increased number of docked vesicles at the active zone, while the overall number of SVs was normal. The active zone length was decreased upon SNX4 depletion,

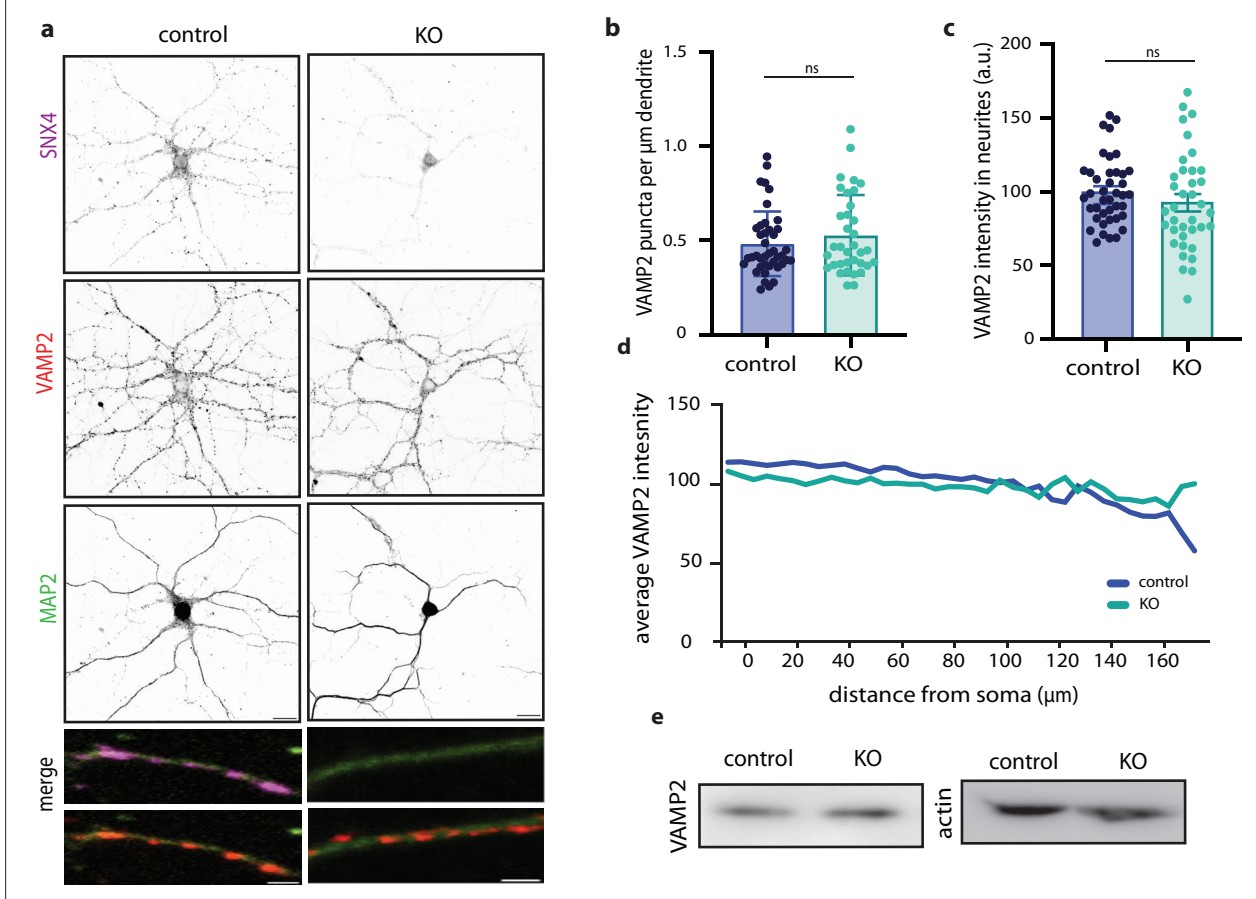

**Figure 4.** SNX4 depletion does not decrease VAMP2 levels and localization. (**a**) Typical example immunocytochemistry images of primary hippocampal neurons stained for SNX4 (magenta), VAMP2 (red), and MAP2 (green) from control and SNX4 cKO mice. Scale bar = 20 μm. Merge zoom-in scale bar = 2 μm. (**b**) VAMP2 puncta per μm dendrite for control and KO. (**c**) VAMP2 intensity measured in dendrites. (**d**) Sholl analysis of average VAMP2 intensity over distance from the soma for control and KO. (**l**) Western blot of SNX4, VAMP2, and actin for control and KO. Data points represent individual neurons, N=6 animals and n=35–41 neurons for VAMP2 data, bar graphs represent mean with SEM.

The online version of this article includes the following source data and figure supplement(s) for figure 4:

**Source data 1.** Original files for western blot analysis displayed in *Figure 4e*.

**Source data 2.** EPS file of the full western blots displayed in *Figure 4e*.

**Figure supplement 1.** Autophagy markers remain unaffected upon SNX4 depletion.

**Figure supplement 1—source data 1.** Original files for western blot analysis displayed in *Figure 4—figure supplement 1d*.

**Figure supplement 1—source data 2.** EPS file of the full western blots displayed in *Figure 4—figure supplement 1d*.

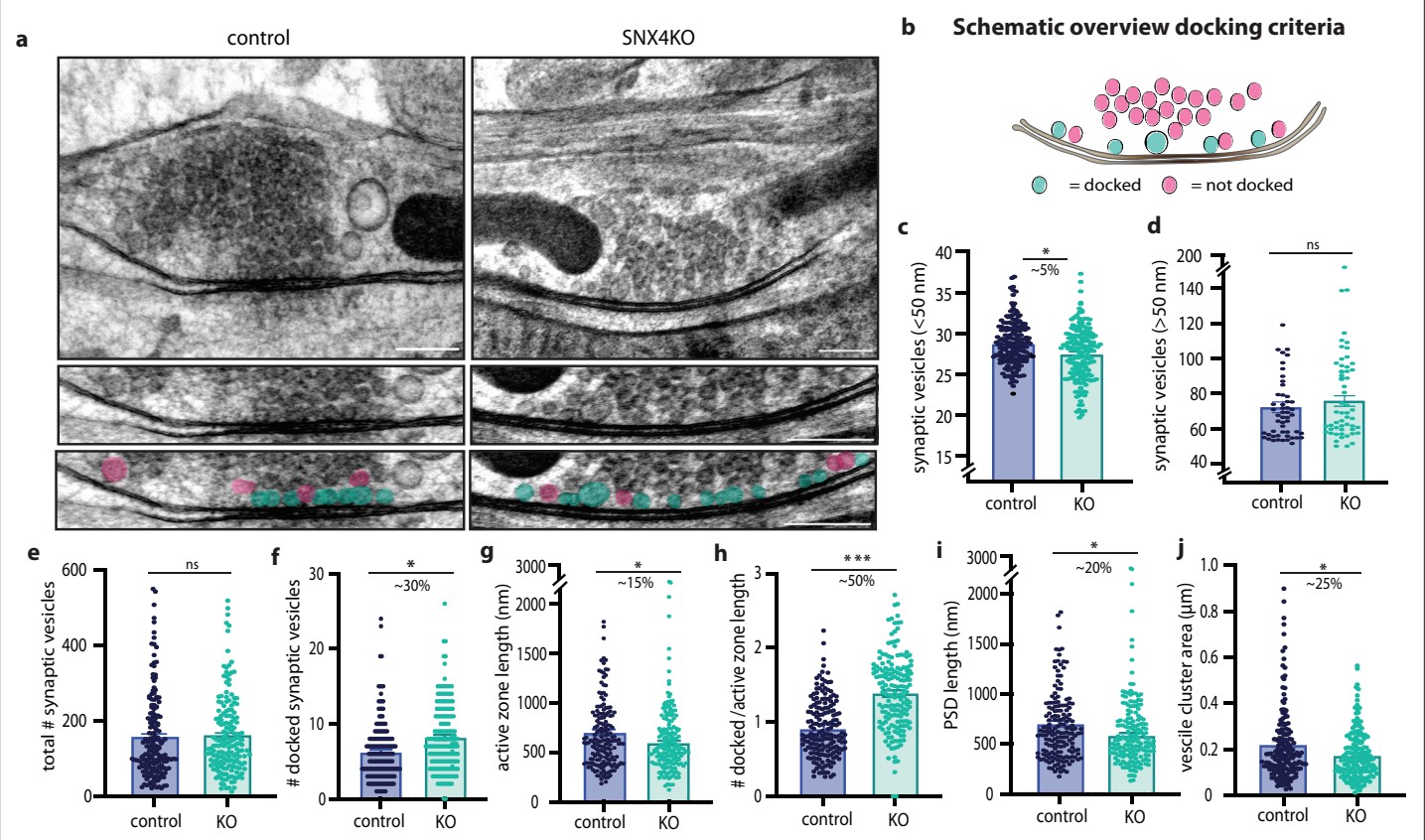

**Figure 5.** SNX4 depletion increases the number of docked vesicles at the active zone. (**a**) Example micrographs of control and SNX4 cKO synapses (scale bar: 200 nm) with zoom-in on the active zone region indicating docked (blue) and non-docked vesicles (pink) (scale bar: 100 nm). (**b**) Schematic overview of docked and undocked vesicles. (**c** and **d**) The average vesicle size, distinguished between synaptic vesicles (<50 nm) and larger clear core vesicles (>50 nm). (**e**) The total amount of synaptic vesicles in the synaptic cloud. (**f**) The number of docked synaptic vesicles in control and SNX4 cKO synapses. (**g**) The active zone length. (**h**) The number of docked vesicles normalized to active zone length. (**i**) The postsynaptic density (PSD) length. (**j**) The area of the vesicle cloud in control and SNX4 cKO synapses. Bar graphs represent mean with SEM with n=174–175 synapses per group of N=3 animals. Data points are individual synapses. ***p<0.0001, **p<0.001 using Mann-Whitney test.

The online version of this article includes the following figure supplement(s) for figure 5:

**Figure supplement 1.** Histogram of vesicle size for control and SNX4KO synapses.

thereby further increasing the density of docked SVs at the release site. Taken together, our data shows that the endosomal sorting protein SNX4 is a negative regulator of SV docking and release. These findings indicate that SNX4 regulates SV recruitment at the active zone.

## Enhanced SV recruitment and increased SV docking explain increased transmission in SNX4 cKO synapses

SNX4 cKO synapses showed a 110% increase in neurotransmission during intense stimulation (*Figure 2f*). A trend toward increased neurotransmission was observed already from the start of the train, and also in single evoked responses (*Figure 2c and d*), but became more prominent (and significant) during sustained stimulation. Despite the variability within all datasets, both trends and the significant effect during sustained stimulation were fully reversed by acute SNX4 expression. The total number of synapses (*Figure 1h*) and fraction of active synapses (*Figure 3g*) was not affected in SNX4 cKO neurons, suggesting an intrasynaptic effect. While approximately twice more vesicles were released during sustained stimulation in the absence of SNX4 (*Figure 2f*), our RRP estimation indicated a normal resting-state RRP (*Figure 2i*). All measures of release probability were normal (normal paired pulse facilitation, *Figure 2e*, and shape of the rundown curve, *Figure 2f*), indicating that an increased release probability is an unlikely explanation for the observed increase

in neurotransmission during sustained stimulation. The mEPSC recordings were also not affected (*Figure 2k–m*), this is consistent with the unaffected RRP estimate and single EPSC size and suggests that the observed increase in neurotransmission is not due to postsynaptic effects. Hence, this suggests that a higher recruitment rate of SVs to the releasable pool (*Figure 2h and j*) in SNX4 cKO synapses is responsible for the increased neurotransmission. While back extrapolation methods for estimating the RRP have their limitations (*Neher, 2015*) and mechanisms of SV recruitment remain not fully understood, it is evident that the involvement of newly recruited vesicles becomes more prominent during high-frequency trains. In line with this, the increased neurotransmission of SNX4 cKO neurons was less prominent in 5 and 10 Hz trains (*Figure 2—figure supplement 1*). Hence, an increased contribution of newly recruited vesicles during sustained stimulation is consistent with the fact that neurotransmission is increased especially during high-frequency stimulation in SNX4 cKO synapses. No effects on neurotransmission were observed when overexpressing SNX4 on a control background, indicating that SNX4 is required but not rate limiting to limit synaptic release during prolonged stimulation.

The increase in neurotransmission during sustained stimulation was also observed with SV fusion reporter SypHy but not with the FM-64 dye. The latter method reports the release of only recycled vesicles, suggesting that these do not contribute significantly to the enhanced release. Finally, examining SV recycling with FM-64 dye indicates that the increased SV recruitment in SNX4 cKO synapses is not driven by an enhanced rate of the SV cycle, as endocytosis and release of recycled SVs were normal in SNX4 cKO synapses.

Ultrastructural characterization of SNX4 cKO synapses revealed a 50% increase in docked vesicles when normalized to active zone length. This increase might contribute to the enhanced SV recruitment and increased neurotransmission observed in SNX4 cKO synapses. If more vesicles are already docked at the start of stimulation, they may on average be recruited faster during train stimulation. SV docking is probably a dynamic equilibrium between various SV pools, modulated by activity (*Kusick et al., 2020*). However, an increase in docked SVs at the active zone also implies, on average, a closer proximity to calcium channels and therefore a higher release probability. However, measures of release probability were unaffected by SNX4 deficiency (as argued above), suggesting a normal channel coupling in SNX4 cKO synapses.

## SNX4 at the crossroad of endosomal sorting and autophagy regulating synaptic output

Increased neurotransmission upon depletion of endosomal genes has been shown by several labs (*Fernandes et al., 2014*; *Soykan et al., 2021*; *Tagliatti et al., 2016*; *Uytterhoeven et al., 2011*). These results are consistent with a postulated hypothesis that endosomes act as sorting stations for SV proteins and control the balance between novel and reused SV proteins that affect SV release and neurotransmission efficiency (*Fernandes et al., 2014*; *Truckenbrodt et al., 2018*; *Uytterhoeven et al., 2011*).

In the presence of SNX4, the recycling process may lead to the reuse of older SV proteins that, as indirect consequence, potentially diminish the efficiency of SV exocytosis. In the absence of SNX4, older SV proteins are less recycled and traffic toward degradation, rendering the synapse with relatively more novel SV proteins, enhancing SV recruitment and fusion. Blocking the degradation pathway with HOPS/ESCRT mutants rescues the observed increased release (*Fernandes et al., 2014*; *Uytterhoeven et al., 2011*). Similarly, Skywalker and ARF6 mutants show increased SV sorting through endosomal intermediates, increasing SV docking and release (*Fernandes et al., 2014*; *Tagliatti et al., 2016*; *Uytterhoeven et al., 2011*). Our data is consistent with this literature, demonstrating enhanced neurotransmission upon inactivation of an endosome sorting gene. That said, SNX4 cKO synapses do not show an accumulation of endosomal intermediates (*Figure 5d*) or decreased SV numbers (*Figure 5e*) as reported for other endosome gene mutants, marking a novel finding in endosomal protein knock-out studies.

Endosomal sorting also mediates the delivery of active zone proteins, where disturbances in these pathways potentially correlate with smaller active zones and SVs, ultrastructural phenotypes were also observed in SNX4 cKO synapses. Previous studies suggest that the removal of SNX4 directs its cargo toward the degradative pathway (*Hettema et al., 2003*; *Traer et al., 2007*). Although the SNX4 cargo, particularly in mammalian neurons, is currently unknown, it is unlikely that VAMP2/syb-2

recycling explains our phenotype since the overall levels and localization of VAMP2/syb-2 remained unaffected upon SNX4 depletion.

Apart from endosome sorting, SNX4 has been reported to regulate several types of autophagy (*Antón et al., 2020*; *Hanley et al., 2021*; *Zhou et al., 2022*). Recent literature has shown that SV release could be regulated by synaptic autophagy (reviewed by *Decet and Verstreken, 2021*). Loss of ATG5 has been shown to result in increased neurotransmission, an effect independent from total SVs pool number (*Kuijpers et al., 2021*). We did not observe changes in total ATG5 levels upon SNX4 depletion, consistent with previous reports (*Antón et al., 2020*), and no difference in basic autophagic flux of P62, contrasting with previous findings in HeLa cells (*Antón et al., 2020*). This suggests that SNX4 may have a specialized role in neurons, that regulates synaptic transmission independent of autophagic flux by a yet-to-be determined mechanism.

## SNX-BAR proteins critical for synapse function

Many of our examined synaptic parameters remained unaltered or showed considerable variability, including our hypothesis that depletion of a synaptic endosomal sorting protein would disrupt synaptic recycling. This suggests the presence of redundant pathways compensating for SNX4 depletion, potentially involving other SNX-BAR complexes or endosomal sorting complexes like ESCPE, retriever, or retromer (*Simonetti et al., 2023*). Particularly SNX-BAR proteins, such as SNX7 and SNX30, are of potential interest for future research due to their ability to form selective complexes with SNX4 (*Antón et al., 2020*). The role of endosome sorting in the SV cycle is increasingly recognized through knock-out studies targeting various endosomal complexes, revealing changes in SV exocytosis, SV endocytosis, or endosome-like vesicle numbers (*Chen et al., 2017*; *Chen et al., 2008*; *Inoshita et al., 2017*; *Tagliatti et al., 2016*). Our data supports this, showing for the first time that a protein from the highly evolutionary conserved SNX-BAR protein family, such as SNX4, regulates SV docking at the active zone and sustained SV release. Taken together, we show that endosomal sorting protein SNX4 regulates SVs recruitment at the active zone. The precise regulation of vesicle availability is crucial for a synapse to effectively coordinate neurotransmission, a process heavily relying on endosome sorting mechanisms.

# Materials and methods

## Laboratory animals and SNX4 cKO generation

Animals were housed and bred according to institutional and Dutch governmental guidelines (DEC-FGA 11-03 and AVD112002017824). The SNX4 conditional KO targeting strategy aimed to achieve mouse SNX4 gene loss of function by flanking exons 3 and 4 with loxP sites to acutely remove these exons by Cre recombinase (*Figure 1a*). In the targeting vector, self-deletion anchor sites flank the neomycin cassette, and diphtheria toxin A fragment is used for negative selection. Mouse genomic fragments containing homology arms were amplified from a BAC clone using high-fidelity Taq DNA polymerase. These fragments were then sequentially assembled into a targeting vector along with the recombination sites and selection markers. Homologously targeted *Snx4*^lox embryonic stem cells (C57BL/6) were injected into blastocysts and implanted into pseudopregnant females to produce germline chimeras, which were mated with inbred C57BL/6 mice. The mouse colony was maintained by homozygous lox/lox mice breeding, with every third or fourth generation mice were backcrossed with C57BL/6 mice. The following primers were used for genotyping: forward primer: TTCTTGAG GGTAACAGAAATCCTTAGTGCC and reverse primer: CTCTGAATACCAGGAGAGTCCACAAGAGC.

## Primary neuronal cultures

Dissociated hippocampal neuron cultures were prepared from P01 SNX4KO mouse embryos. Hippocampi dissected free of meninges in Hanks' balanced salt solution (Sigma, H9394) supplemented with 10 mM HEPES (Gibco, 15630-056). The hippocampi were isolated from the tissue and digested with 0.25% trypsin (Gibco, 15090-046) in Hanks' HEPES for 20 min at 37°C. Hippocampi were washed three times with Hanks' HEPES and triturated with fire-polished glass pipettes. Neurons were counted and plated in Neurobasal medium (Gibco, 21103-049) supplemented with 2% B-27 (Gibco, 17504-044), 1.8% HEPES, 0.25% GlutaMAX (Gibco, 35050-038), and 0.1% penicillin-streptomycin (Gibco, 15140-122). Hippocampal neurons were plated in 12-well plates at a density of 25,000 cells per well on

18 mm glass coverslips containing rat glia or 50,000 cells on coverslips containing 0.01% Poly-ornithin (Sigma P4957) and 2.5 µg/ml laminin coating (Sigma L2020) solution overnight at room temperature. Neuronal cultures were kept in supplemented Neurobasal at 37°C and 5% $CO_2$ until DIV21.

## Constructs and lentiviral infection

Neuronal cultures were infected with EGFP-Cre and EGFP-control (delta-Cre) at DIV0 to generate SNX4KO. All constructs were cloned into lentiviral vectors under the synapsin promoter, and lentiviral particles were generated following the protocol previously described in *Naldini et al., 1996*. The presence of an EGFP nuclear localization sequence allowed identification of tranduced cells (*Kaeser et al., 2011*). For SV release experiments, neurons were infected with synaptophysin-pHluorin lentivirus at DIV7. Titration of lentiviral particle batches was performed by assessment of number of fluorescent cells upon infection to ensure 100% infection efficiency. For rescue experiments and SNX4 overexpression experiments a lentiviral construct of mouse SNX4 was generated with an ECFP fluorescent tag together with an 2A cleavage site in between in an FUW backbone (pECFP-T2A-SNX4 [*Mus musculus*]-FUW).

## Western blot

Cortical neurons at DIV21 were washed with phosphate-buffered saline (PBS) and lysed in Laemmli sample buffer consisting of 2% SDS (VWR Chemicals, M107), 10% glycerol (Merck, 818709), 0.26 M β-mercaptoethanol (Sigma, M3148), 60 mM Tris-HCl (Serva, 37180) pH 6.8, and 0.01% Bromophenol blue (AppliChem, A3640). Samples of 450,000 neurons run in SDS-PAGE (10% 1 mm acrylamide gel with 2,2,2-trichloroethanol) using standard SDS-PAGE technique and transferred into polyvinylidene fluoride membranes (Bio-Rad) (1 hr, 0.25 mA, 4°C). Blots were blocked and incubated with primary antibodies (4°C overnight) in PBS containing 0.1% Tween-20 (Sigma, P2287) (PBS-T) and 5% milk. Primary antibodies included polyclonal rabbit SNX4 (1:500; SySy, Cat. No. 392-003), monoclonal mouse actin (1:10,000; Chemicon, Cat. No. MAB1501), monoclonal mouse VAMP2/synaptobrevin2 (1:2000; SySy, CL 69.1), and polyclonal rabbit BACE-1 (1:500; Abcam ab20770). After washing with PBS-T, blots were incubated with secondary horseradish peroxidase-labeled antibodies (1:10,000; Sigma) in PBS-T and 5% milk (1 hr, room temperature). After washing with PBS-T, blots were developed with SuperSignal West Femto substrate (Thermo Scientific, 11859290) for 2 min. Chemiluminescence-labeled proteins were visualized with the Odyssey Imaging System (LI-COR) for appropriate times and analyzed with Image Studio 5.2 software (LI-COR) or ImageJ. A fixed region of interest (ROI) was used to measure the average intensity, which was then normalized to (1) the loading control (housekeeping genes actin or GAPDH) and (2) the control within each independent experiment (N) to account for weekly variations.

## Immunocytochemistry

Neurons at DIV21 were fixed in 3.7% formaldehyde (Electron Microscopy Sciences, 15680) in PBS, pH 7.4, for 25 min at room temperature. After washing with PBS, cells were permeabilized for 5 min with 0.5% Triton X-100 (Fisher Chemical, T/3751/08)-PBS and incubated for 30 min with PBS containing 2% normal goat serum (Gibco, 16210-072) and 0.1% Triton X-100 to block nonspecific binding. Incubations with primary antibodies were performed overnight at 4°C. After PBS washing steps incubations with secondary antibodies followed for 1 hr at room temperature. Primary antibodies included polyclonal rabbit SNX4 (1:500; SySy, Cat. No. 392 003), monoclonal mouse VAMP2/synaptobrevin2 (1:1000; SySy, CL 69.1), and polyclonal chicken MAP2 (1:300; Abcam ab5392) and Alexa Fluor-conjugated secondary antibodies (1:1000; Invitrogen). Coverslips were mounted with Dabco-Mowiol 4-88 (Sigma, 81381) and confocal microscope (Nikon Eclipse Ti) using a ×40 oil immersion objective (NA = 1.3) was used to examine the samples. Laser settings were kept constant within experimental conditions.

## Electrophysiological recordings

Autaptic cultures of *SNX4* null neurons were grown for 21–25 days before measuring. Whole-cell voltage-clamp recordings (Vm = −70 mV) were performed at room temperature with borosilicate glass pipettes (2.5–4.5 mOhm) filled with 136 mM KCl, 17.8 mM HEPES, 0.6 mM $MgCl_2$, 4 mM ATP-Mg, 0.3 mM GTP-Na, 12 mM phosphocreatine dipotassium salt, 45.3 mM phosphocreatine kinase, and 1 mM EGTA (pH 7.30). External solution contained the following (in mM): 10 HEPES, 10 glucose,

140 NaCl, 2.4 KCl, 4 MgCl$_2$, and 4 CaCl$_2$ (pH = 7.30, 300 mOsmol). Recording was acquired with an MultiClamp 700B amplifier (Molecular Devices), Digidata 1550B, and Clampex 9.0 software (Molecular Devices). After whole-cell mode was established, only cells with an access resistance of <15 MΩ and leak current of <300 pA were accepted for analysis. Series resistance compensation was between 70% and 80%. Access resistance was measured before and after recordings. Recordings with a >20% increase in access resistance were excluded from analysis, as well as recordings that showed voltage-clamp escape or clipping. GABAergic recordings were identified based on their postsynaptic decay kinetics and excluded. EPSCs were elicited by a 1 ms depolarization to 30 mV. Offline analysis was performed with MATLAB R2019a (Mathworks) using custom-written software routines (viewEPSC, downloaded from user vhuson on GitHub on September 13, 2023; *Huson, 2019*). RRP and slope estimates were obtained by a linear fit of cumulative total charge plot from pulse 80 to 100 for each individual cell with y-intercept for RRP estimate and slope as recruitment rate estimate (*Schneggen-burger et al., 1999*).

## Live-cell imaging

Live imaging experiments were performed at DIV21 on a custom-build imaging microscope (IX81; Olympus) with an MT20 light source (Olympus) with a ×40 oil objective (NA = 1.3) appropriate filter sets (Semrock, Rochester, NY, USA) and an EM charge-coupled device camera (C9100-02; Hamamatsu Photonics, Japan). Xcellence RT imaging software (Olympus) controlled the microscope and image acquisition. Coverslips were perfused with Tyrode's solution (119 mM NaCl, 2.5 mM KCl, 2 mM CaCl$_2$-2H$_2$O, 2 mM MgCl$_2$-6H$_2$O, 25 mM HEPES, and 30 mM glucose H$_2$O, pH 7.4, mOsmol 280). For SV release experiments, time-lapse (2 Hz) recordings consisted of 30 s baseline recordings followed by two electrical field stimulations with 60 s recovery time in between and a final stimulation 10 s of NH4-Tyrode's perfusion (2 mM CaCl$_2$, 2.5 mM KCl, 119 mM NaCl, 2 mM MgCl$_2$, 30 mM glucose, 25 mM HEPES, 50 mN NH$_4$Cl at pH 7.4). Electrical field stimulation was applied via parallel platinum electrodes powered by a stimulus isolator (WPI A385) delivering 30 mA, 1 ms pulses, regulated by a Master-8 pulse generator (AMPI) providing 100 APs at 40 Hz with a 0.5 s interval. Glass capillaries close to the isolated neuron were used for chemical stimulation. For the FM4-64 experiments, neurons were pre-imaging incubated in high potassium Tyrode's for 10 min at 37°C (61.5 mM NaCl, 60 mM KCl, 2 mM CaCl$_2$2H$_2$O, 2 mM MgCl$_2$ 6H$_2$O, 25 mM HEPES, and 30 mM glucose H$_2$O, pH 7.4, mOsmol 280).

## Imaging analysis

For live-cell imaging analysis, time-lapse recordings stacks were used to analyze SyPhy and FM4-64 fusion events (2 Hz, 1000×1000 pixels). In ImageJ fusion events were selected manually and fluorescence traces were measured in a circular 3×3 pixel ROI (0.6×0.6 μm$^2$), somatic events not included. ROI intensity measures of SyPhy puncta were exported to a custom-built MATLAB program for semi-automated detection of fusion events and duration (*Moro et al., 2021*). Fluorescence traces from the ROI were expressed as fluorescence change (deltaF) compared to initial fluorescence (F0), with the average of frames 1–10 for determining F0. Fusion events were automatically detected when displaying a sudden increase of two standard deviations above F0. SV pool numbers and synapses were estimated using MAP2 mask in automated image analysis software SynD (*Schmitz et al., 2011*). Active synapses were determined by 3× SD over baseline. Synapse detection settings were optimized to measure SyPhy and VAMP2/syb-2 puncta signal and kept constant within the same dataset. For FM4-64 puncta, fluorescence traces were expressed as change (DF) compared to baseline fluorescence (F0) and the average of the last 10 frames was used to determine the maximum FM4-64 release. For fixed-cell imaging analysis, VAMP2 puncta number within MAP2 mask were analyzed using automated SynD software (see above). For Sholl analysis, dendrite mask based on MAP2 staining was used to either determine the number of dendritic branches per shell or average VAMP2 intensity within the neurite mask.

## Cryofixation electron microscopy

Dissociated hippocampal neurons (5000 k/well) from SNX4 cKO mice were plated on pre-grown cultures of rat glia on sapphire disks (Wohlwend GmbH) to form micro-networks of 2–10 neurons per sapphire disk. Prior to cell culture, sapphire disks were etched for 30 min in 60% sulfuric acid, washed,

incubated in 3 M KOH overnight, washed and dried before carbon coating and subsequent baking at 180°C for 2 hr. Sapphire discs were coated by a mixture of 0.1 mg/ml poly-d-lysine (Sigma), 0.7 mg/ml rat tail collagen (BD Biosciences), and 10 mM acetic acid (Sigma) and placed in an agarose-coated 12-well plate to form glia monolayer islands selectively on sapphire disks. The sapphire disks were cryofixed in an EM-PACT2 (Leica Microsystems) high-pressure freezer in 5% trehalose/10% BSA in 0.05 M phosphate buffer pH 7.4 320 mOsm cryoprotectant. Frozen samples were postfixed in 1% $OsO_4$/5% saturated K4Fe(CN)6 in $H_2O$ in acetone at −90°C for 74 hr and brought to 0°C at 5°C/hr. After a couple washes with ice-cold acetone, the sapphire disks were washed with propylene oxide and with an increasing Epon concentration series. The samples were embedded in fresh Epon overnight and left to polymerize at 65°C for 48 hr. Sapphire disks were separated from the Epon by dipping the samples in boiling water and liquid nitrogen and regions with micro-networks were selected by light microscopy. These regions were cut out and mounted on pre-polymerized Epon blocks for ultrathin sectioning. Ultrathin sections (80 nm) were cut parallel to the sapphire disk, collected on single-slot, formvar-coated copper grids, and stained in uranyl acetate and lead citrate. Hippocampal synapses were randomly selected at low magnification using an electron microscope (JEOL1010), and the conditions were blinded. For each condition, the number of docked SVs, total SV number, and active zone length were measured on digital images taken at 80,000-fold magnification using custom-written semiautomatic image analysis software running in MATLAB (Mathworks). For all morphological analyses the following requirements were set: clearly recognizable synapses with intact synaptic plasma membranes with a recognizable pre- and postsynaptic area and defined SV membranes. SVs were defined as docked if there was no distance visible between the SV membrane and the active zone membrane (pixel size 0.6505 nm).

## Statistics

Data are shown as mean ± SEM. Data that violated both normality (Shapiro-Wilk) and homogeneity (Levene test) was tested with the two-sided Mann-Whitney U test. Data exhibiting a normal distribution yet significantly altered variation were tested with Welch's t-test. When assumptions of homogeneity and normality were met, data were tested with Student's t-test or one-way ANOVA with Tukey's multiple comparisons test. For the analysis of electron microscopy data, electrophysiology data and live-cell imaging (pHluorin and FM dye), the intraclass correlation coefficient (ICC) was calculated for different variables applicable to each experiment such as culture batch, sapphire, or synapse originating from the same neuron. If ICC was close to 0.1 or above, indicating that a considerable portion of the total variance can be attributed to, e.g., culture week, multilevel analysis was performed to accommodate nested data (*Aarts et al., 2014*). For electrophysiology data, we excluded outliers beyond mean ± 3× SD. Bar graphs were plotted as the mean with SEM, where individual data points represent fields of view, neurons, or synapses (EM dataset).

## Acknowledgements

The authors thank Dr. Sander Groffen for help with the conditional knock-out mouse design, Joke Wortel for housing and breeding the mice, Desiree Schut for helping with neuronal cultures, Robbert Zalm for cloning and lentiviral production, Marien P Dekker for assistance with EM analysis. We thank Jill Gerritsen for electrophysiology data analysis. We would like to thank Elise van Bree and Sonia Vazquez-Sanchez for their pioneering work on synaptic release in SNX4 shRNA models. The van Weering lab is supported by Alzheimer Nederland (WE.03-2019-06).

## Additional information

### Funding

| Funder | Grant reference number | Author |
| --- | --- | --- |
| Alzheimer Nederland VU University | WE.03-2019-06 | Jan RT van Weering |

| Funder | Grant reference number | Author |
|---|---|---|

The funders had no role in study design, data collection and interpretation, or the decision to submit the work for publication.

## Author contributions

Josse Poppinga, Conceptualization, Data curation, Software, Formal analysis, Validation, Investigation, Visualization, Methodology, Writing – original draft, Project administration, Writing – review and editing; Nolan J Barrett, Data curation, Formal analysis, Investigation, Methodology, Project administration; L Niels Cornelisse, Conceptualization, Resources, Data curation, Supervision, Writing – review and editing; Matthijs Verhage, Conceptualization, Resources, Supervision, Funding acquisition, Writing – original draft, Writing – review and editing; Jan RT van Weering, Conceptualization, Resources, Data curation, Supervision, Funding acquisition, Investigation, Methodology, Writing – original draft, Project administration, Writing – review and editing

## Author ORCIDs

Josse Poppinga ⓘ https://orcid.org/0009-0002-6809-3832
Jan RT van Weering ⓘ https://orcid.org/0000-0001-5259-4945

## Ethics

Animals were housed and bred according to institutional and Dutch governmental guidelines (DEC-FGA 11-03 and AVD112002017824).

Reviewer #1 (Public review): https://doi.org/10.7554/eLife.97910.3.sa1
Reviewer #2 (Public review): https://doi.org/10.7554/eLife.97910.3.sa2
Reviewer #3 (Public review): https://doi.org/10.7554/eLife.97910.3.sa3
Author response https://doi.org/10.7554/eLife.97910.3.sa4

# Additional files

## Supplementary files

• Supplementary file 1. Statistics overview per figure. Overview of experiments per figure with N number, mean, median, SEM, SD, performed statistical test, and p-value.

• MDAR checklist

## Data availability

All supplemental data, used to create graphs and its raw data, is made available on Dataverse. https://doi.org/10.34894/RMKROK.

The following dataset was generated:

| Author(s) | Year | Dataset title | Dataset URL | Database and Identifier |
|---|---|---|---|---|
| Poppinga J, Barrett NJ, Cornelisse LN, Verhage M, van Weering JRT | 2024 | Raw imaging and ephys data SNX4 cKO per figure | https://doi.org/10.34894/RMKROK | DataverseNL, 10.34894/RMKROK |

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
