## [Editor Report · eLife Assessment]

This **important** study presents a series of results aimed at uncovering the involvement of the endosomal sorting protein SNX4 in neurotransmitter release. While the evidence supporting the conclusions is **solid**, the molecular mechanisms remain unclear. This paper will be of interest to cell biologists and neurobiologists.

---

## [Referee Report · Reviewer #1 (Public review)]

Summary:

In the work Josse Poppinga and collaborators addressed the synaptic function of Sortin-Nexin 4 (SNX4). Employing a newly-developed in vitro KO model, with live imaging experiments, electrophysiological recordings and ultrastructural analysis, the authors evaluate modifications in synaptic morphology and function upon loss of SNX4. The data demonstrate increased neurotransmitter release and alteration in synapse ultrastructure with higher number of docked vesicles and shorter AZ. The evaluation of presynaptic function of SNX4 is of relevance and tackles an open and yet unresolved question in the field of presynaptic physiology.

Strengths:

The sequential characterization of the cellular model is nicely conducted, and the different techniques employed are appropriate for the morpho-functional analysis of the synaptic phenotype and the derived conclusions on SNX4 function at presynaptic site. The authors succeeded in presenting a novel in vitro model that results in chronic deletion of SNX4 in neurons. A convincing sequence of experimental techniques are applied to the model to unravel the role of SNX4, whose functions in neuronal cells and at synapses are largely unknown. The understanding of the role of endosomal sorting at presynaptic site is relevant and of high interest in the field of synaptic physiology and on the pathophysiology of the many described synaptopathies that broadly result in loss of synaptic fidelity and quality control at release sites.

Weaknesses:

The flow of the data presentation is mostly descriptive with several consistent morphological and functional modifications upon SNX loss. The paper would benefit from a wider characterization that would allow to address the physiological roles of SNX4 at synaptic site and speculate on the underlying molecular mechanisms. The novel experiments on autophagy progression as well as spontaneous neurotransmission are well conducted, although do not assist for the explanation of the molecular mechanism underneath.

Comments on revisions:

Other implementations in the revised version are quite limited and would benefit from a more detailed presentation and description. i.e.: Sholl analysis in the new figure 1h, is presented with no definition of number of cells employed and standard deviations of the replication. The "simil" Sholl analysis performed on VAMP2 is still puzzling and some explanations on the reason for the constant value of VAMP2 fluorescent signal from less than 0 to 160 µm from the cell body is to be added. How is the increased number of active synapses explained? How is this related to shorter AZ and higher number of docked vesicles?

---

## [Referee Report · Reviewer #2 (Public review)]

Summary:

SNX4 is thought to mediate recycling from endosomes back to the plasma membrane in cells. In this study, the authors demonstrate the increases in the amounts of transmitter release and the number of docked vesicles by combining genetics, electrophysiology and EM. They failed to find evidence for its role in synaptic vesicle cycling and endocytosis, which may be intuitively closer to the endosome function.

Strengths:

The electrophysiological data and EM data are in principle, convincing, though there are several issues in the study.

Weaknesses:

It is unclear why the increase in the amounts of transmitter release and docked vesicles happened in the SNX4 KO mice. In other words, it is unclear how the endosomal sorting proteins in the end regulate or are connected to presynaptic, particularly the active zone function.

Comments on revisions:

I am fine with revision in principle. the authors have addressed my concerns.

---

## [Referee Report · Reviewer #3 (Public review)]

Summary:

The study aims to determine whether the endosomal protein SNX4 performs a role in neurotransmitter release and synaptic vesicle recycling. The authors exploited a newly generated conditional knockout mouse to allow them to interrogate SNX4 function. A series of basic parameters were assessed, with an observed impact on neurotransmitter release and active zone morphology. The work is interesting, however as things currently stand, the work is descriptive with little mechanistic insight. There are a number of places where some of the conclusions require further validation.

Strengths:

The strengths of the work are the state-of-the-art methods to monitor presynaptic function.

Weaknesses:

The weaknesses are the fact that the work is largely descriptive, with no mechanistic insight into the role of SNX4.

Comments on revisions:

The authors have addressed a couple of the more major concerns with the manuscript, however many of the original weaknesses remain. The primary weakness being the lack of mechanism. It is disappointing that real-time VAMP2 trafficking was not investigated, and the authors justification as to why the experiment was not performed was not convincing (especially since this is the approach that all other groups employ to examine SV cargo trafficking). In a number of instances "contractual constraints" are referred to as an explanation for not performing additional experiments. It was unclear whether this refers to licencing issues with the mouse line or the lack of personnel to perform the work. Regardless it still leaves this work as somewhat incomplete.

---

## [Author Response]

The following is the authors’ response to the original reviews.

**Public Reviews:**

**Reviewer #1 (Public Review):**
Summary:In the work: "Endosomal sorting protein SNX4 limits synaptic vesicle docking and release" Josse Poppinga and collaborators addressed the synaptic function of Sortin-Nexin 4 (SNX4). Employing a newly developed in vitro KO model, with live imaging experiments, electrophysiological recordings, and ultrastructural analysis, the authors evaluate modifications in synaptic morphology and function upon loss of SNX4. The data demonstrate increased neurotransmitter release and alteration in synapse ultrastructure with a higher number of docked vesicles and shorter AZ. The evaluation of the presynaptic function of SNX4 is of relevance and tackles an open and yet unresolved question in the field of presynaptic physiology.Strengths:The sequential characterization of the cellular model is nicely conducted and the different techniques employed are appropriate for the morpho-functional analysis of the synaptic phenotype and the derived conclusions on SNX4 function at presynaptic site. The authors succeeded in presenting a novel in vitro model that resulted in chronical deletion of SNX4 in neurons. A convincing sequence of experimental techniques is applied to the model to unravel the role of SNX4, whose functions in neuronal cells and at synapses are largely unknown. The understanding of the role of endosomal sorting at the presynaptic site is relevant and of high interest in the field of synaptic physiology and in the pathophysiology of the many described synaptopathies that broadly result in loss of synaptic fidelity and quality control at release sites.

We thank the reviewer for their positive evaluation of our manuscript.

Weaknesses:The flow of the data presentation is mostly descriptive with several consistent morphological and functional modifications upon SNX loss. The paper would benefit from a wider characterization that would allow us to address the physiological roles of SNX4 at the synaptic site and speculate on the underlying molecular mechanisms. In addition, due to the described role of SNX4 in autophagy and the high interest in the regulation of synaptic autophagy in the field of synaptic physiology, an initial evaluation of the autophagy phenotype in the neuronal SNX4KO model is important, and not to be only restricted to the discussion section.

We thank the reviewer for their suggestions and agree that broader characterization would help us speculate on the underlying mechanism. To address this, we have conducted additional independent experiments investigating the role of SNX4 in neuronal autophagy, as suggested by this reviewer. These experiments are now included in the main figures and are no longer limited to the discussion section. Please see the detailed responses to this reviewer's recommendations below.

**Reviewer #2 (Public Review):**
Summary:SNX4 is thought to mediate recycling from endosomes back to the plasma membrane in cells. In this study, the authors demonstrate the increases in the amounts of transmitter release and the number of docked vesicles by combining genetics, electrophysiology, and EM. They failed to find evidence for its role in synaptic vesicle cycling and endocytosis, which may be intuitively closer to the endosome function.Strengths:The electrophysiological data and EM data are in principle, convincing, though there are several issues in the study.

We thank the reviewer for their positive evaluation of our manuscript.

Weaknesses:It is unclear why the increase in the amounts of transmitter release and docked vesicles happened in the SNX4 KO mice. In other words, it is unclear how the endosomal sorting proteins in the end regulate or are connected to presynaptic, particularly the active zone function.

We thank the reviewer for their suggestions and agree that further characterization would help to understand how endosomal sorting proteins regulate presynaptic neurotransmission. We have now added extra data on electrophysiological recordings clarifying SNX4’s role in the synapse. Please see the detailed responses to this reviewer's recommendations below.

**Reviewer #3 (Public Review):**
Summary:The study aims to determine whether the endosomal protein SNX4 performs a role in neurotransmitter release and synaptic vesicle recycling. The authors exploited a newly generated conditional knockout mouse to allow them to interrogate the SNX4 function. A series of basic parameters were assessed, with an observed impact on neurotransmitter release and active zone morphology. The work is interesting, however as things currently stand, the work is descriptive with little mechanistic insight. There are a number of places where the data appear to be a little preliminary, and some of the conclusions require further validation.Strengths:The strengths of the work are the state-of-the-art methods to monitor presynaptic function.

We thank the reviewers for their positive evaluation of our manuscript.

Weaknesses:The weaknesses are the fact that the work is largely descriptive, with no mechanistic insight into the role of SNX4. Further weaknesses are the absence of controls in some experiments and the design of specific experiments.

We thank the reviewer for their suggestions and agree that addition of extra control groups and experiments would strengthen interpretation of the observed phenotype. To address this, we have now performed experiments to investigate the miniature excitatory postsynaptic currents and added extra control groups such as overexpression of SNX4 on control background. In addition, we assessed SNX4-mediated neuronal autophagy as a potential molecular mechanism by which SNX4 affects synaptic output. Please see the detailed responses to this reviewers’ recommendations below.

**Recommendations for the authors:**

**Reviewer #1 (Recommendations For The Authors):**
(1) The characterization of the neurite outgrowth presented in Figure 1 is a necessary starting point for the characterization of the model and the interpretation of the following data. Being the analysis conducted at 21 DIV, a significant portion of the neurite tree is out of the analyzed field. Adding sholl analysis will better indicate the complexity of the that appears to be influenced by SNX4 loss in the representative images shown in Figure 1f.

We fully agree and have now performed a Sholl analysis of dendrite branches to investigate dendritic complexity. (Figure 1(i), page 2-3, line 86-88). SNX4 depletion does not affect dendrite length or dendrite branching.

(2) Analogously, the characterization of synapse number is of relevance for the interpretation of the data. For a better flow of the data, Figure 4 might be presented as Figure 2 (without the repetition of panel h in Figure 1). An explanation of how VAMP2 puncta are processed is necessary in the method section. A double labelling with a postsynaptic marker would allow trafficking organelles to be distinguished from mature synaptic contacts. Indeed, the analysis of VAMP2 intensity along neurite in mature 21DIV neurons should reveal peaks in the intensity profile that represent synaptic contacts. For unexplained reasons, the profile is rather flat in the two experimental groups. Focusing on axonal branches will surely result in a peaked profile for VAMP2 labelling.

We fully agree that the characterization of synapses is relevant for the interpretation of the data. We have now added a section in our Material and Methods how the VAMP2 puncta are processed (p14 line 517-520). Instead of labeling mature synapses using double labeling of VAMP2 and PSD95, we analyzed the number of active synapses in live neurons using SypHy (Fig. 3g). The reviewer is correct that the VAMP2 data presented in Fig 1I and Fig 4 is part of the same dataset and we have clarified this in the figure legend. In Fig 1I only the total number of VAMP2 puncta is plotted as a marker for synapse number, while in Fig 4 we assess VAMP2 as potential SNX4 sorting cargo (Ma et al., 2017). Because of these different aims, we prefer to keep the figures separate. The analysis of VAMP2 intensity along the distance of the soma is a Sholl analysis (Fig. 4d), represents the average VAMP2 intensity over distance from the soma of 35-41 neurons per group. In contrast to a line scan of a single neurite, this average profile lacks the peaks of individual synapses.

(3) Miniature excitatory postsynaptic currents recordings would strengthen the synaptic characterization and complement the electrophysiological recordings shown in Figure 2. Analyzing frequency and amplitude parameters would complement the data on the number of synaptic connections defined by the pre and postsynaptic colocalization puncta as suggested above and may support the data shown in Figure 3 g that suggests a decreased number of active synapses in SNX4-KO cells.

We fully agree that the characterization of miniature excitatory postsynaptic currents would strengthen the synaptic characterization and complement the other electrophysiological data. Therefore, we have now added additional experiments showing the mEPSCs (Fig. 2k-m, page 4) in SNX4 cKO neurons versus control. This data shows that the amplitude and frequency of spontaneous miniature EPSCs (mEPSCs) were not affected upon SNX4 depletion, consistent with a normal first evoked EPSC and RRP estimate. Furthermore, these data suggest that it is unlikely that the observed increase in neurotransmission is due to post-synaptic effects.

(4) Recordings on the first evoked response shown in Figure 2 b and quantified in Figures c and d suggest that SNX4 overexpression per se exerts some effect on the Amplitude and the Charge of the first evoked response. This is also evident in the supplementary Figure 2 with lower frequency trains. An additional experimental group, namely control+SNX4 is needed for the correct interpretation of the observed phenotype. The possibility that SNX4 per se exerts an effect on evoked transmission could be discussed in terms of putative mechanisms and interactions.

We thank the reviewer for their suggestion and agree that an additional experimental group (control + SNX4) would strengthen interpretation of the observed phenotype. We have now added a new experimental condition with overexpression of SNX4 on a control background (Supplementary Fig. 3, page 20). This data shows that the amplitude and charge of the first evoked response were not affected in control + SNX4 neurons compared to control, and no differences were detected in the response to the 40 Hz stimulation train (Supplementary Fig. 3a-e). Together, these data suggest that SNX4 overexpression in itself does not affect the neurotransmission protocols studied in SNX4 cKO experiments.

(5) To correctly interpret the SyPhy experiments and exclude an effect of SNX silencing on SV recycling, it is suggested to repeat the experiments shown in Figure 3 in the absence and in the presence of bafilomycin. Indeed, the quantifications shown in Figure 3 d and f do not represent "release fraction" as stated (lines 139/140) but they rather refer to an average difference between release fraction and recovered fraction. With the use of bafilomycin, the comparison of the deltaFmax/deltaFNH4Cl with and without bafilomycin would enable the release fraction to be correctly evaluated and compared.

We appreciate the reviewer’s suggestion and agree on the importance of considering the impact of SV recycling when evaluating the released fraction. We agree that the presence of bafilomycin is critical to isolate the released component during stimulation. We have now rephrased this conclusion. To assess synaptic recycling in these assays, bafilomycin in not critically required and we show by multiple independent experiments, including SypHy and FM64 dye assays, that SV recycling is either not affected or the effect is too small to be detected by these methods.

(6) In the ultrastructural analysis, additional quantifications are needed to exclude the accumulation of endosome-like structures. It is not clear if, in the evaluation of total SV number (Figure 5e), the authors counted all vesicles or vesicles < 50nm. This has to be explained and additional quantification of # of SV < 50nm and # SV > 50nm is informative, taking into account the endosomal nature of SNX4. Indeed, although the average size of SV is not changed (fig. 5 d), the density of "bigger vesicle" may result from endosomal-like structure accumulation. An additional suggested quantification is on vesicle # SV > 80nm as previously reported in the cited references dealing with endosomal proteins and presynaptic morphology.

We fully agree that the characterization of vesicle size is important and that it was not clearly stated which vesicles were included in the total number of SV (Fig. 5e). We have now added this to the figure description. We have also added a histogram that contains the vesicle numbers of different bin sizes for SNX4 cKO synapses and control synapses (Supplementary Fig. 4, page 21) including # SVs > 80nm. (Whilst it seems that there are more “bigger” vesicles in the KO, further analysis revealed that this is mostly driven by one experiment and this effect is not consistent.)

(7) Due to the high scientific interest in presynaptic autophagy for SV recycling and degradation, and the paucity of experimental work assessing the proteins involved, an initial evaluation of the neuronal autophagy process (by western blot analysis and immunocytochemistry) for the characterization of the model will better support the paragraph in the discussion (lines 314-322) and contribute to future work in the field. Although very rare, autophagosomes quantification at presynaptic sites can also be performed from the already acquired images. A double membrane structure with the material inside is evident in the representative control image presented!

We appreciate the reviewer’s suggestion and agree that presynaptic autophagy is an interesting potential mechanism that would elaborate our current working model. To address the reviewers’ suggestion, we added multiple independent experiments to investigate basal autophagy markers such as ATG5 using western blot analysis, characterization of p62 levels using immunohistochemistry and performed additional morphometric analysis on the electron microscopy data (Supplementary Fig. 5). In SNX4 cKO neurons, there was no significant difference in P62 puncta numbers or P62 somatic intensity under basal conditions or after blocking autophagic P62 degradation by bafilomycin treatment, suggesting that autophagic flux remains normal. Also, no changes in total ATG5 protein levels were observed and ultrastructural analysis revealed no differences in the total number of autophagosomes. Collectively, these data indicate that SNX4 depletion does not impact the basal autophagic flux, ATG5 protein levels, or the number of autophagosomes.

Minor points:(1) Dorrbaun et al. 2018 is missing from the reference list. In the legend to figure 1 there is an incorrect reference to Figure 6, rather than Figure 4.

We have now adjusted the figure legend and added the reference (page 16, line 604).

(2) Information on the construct employed for the rescue is missing. Is it a fluorescent tag construct? Representative images of the three autaptic neurons (control, KO, KO+SNX4) would nicely complement data presentation in Figure 2.

We have now elaborated on this in material and methods section (p12, line 418-421). Unfortunately, we did not obtain pictures of autaptic neurons used for electrophysiology experiments.

**Reviewer #2 (Recommendations For The Authors):**
(1) Figure 2d and f are somewhat inconsistent. Total charges for the 1st EPSCs differ almost 2-fold in the same condition.

We appreciate the reviewer’s concern. The average EPSCs charge of the first evoked was 89, 122 and 57 pC for control, KO and rescued neurons respectfully. The average charge of the first pulse of 40Hz train was 41,58 and 32 pC for control, KO and rescued neurons respectfully, which is roughly 50% of the naïve response of the same cells. These trains were recorded after 2 or 3 other stimulation paradigms, which can have affected the total charge released in the 40Hz train. That said, the proportional difference between groups is high comparable, with a 37% increased average charge released in SNX4 cKO compared to control in the naïve response and 41% increased response in the first response of the 40 Hz train, and rescued cells show a 53% reduction in average released charge compared to control in the naïve response compared to a 44% reduction in the first response of the 40 Hz train. Although the absolute values differ between these readouts, we conclude that the biological comparison between groups is consistent.

(2) Figure 2h. This type of analysis has a drawback. See Neher (2015) for the problems associated with this analysis.

We fully agree with the reviewer’s comment. As noted in our discussion (page 9 line 285), while this analysis has its limitations, it can still provide an indication of the ready releasable pool.

(3) The EPSC phenotype may be due to postsynaptic effects. This should be excluded by additional experiments (mEPSC analysis) or further clarification.

We fully agree that the characterization of miniature excitatory postsynaptic currents recording would strengthen the synaptic characterization and complement the electrophysiological recordings. Therefore, we have now added additional experiments showing the mEPSCs (Fig. 2k-m) in SNX4 cKO neurons versus control. This data shows that the amplitude and frequency of spontaneous miniature EPSCs (mEPSCs) were not affected upon SNX4 depletion, suggesting that it is unlikely that the observed increase in neurotransmission is due to post-synaptic effects.

(4) The increased number of docked vesicles observed in EM and the increased slope (vesicle recruitment, Figure 2h) are not consistent with each other. Maybe the definition of docked vesicles is unclear in this version of the manuscript.

As noted in our material & methods (page 15, line 547-548), SVs were defined as docked if there was no distance visible between the SV membrane and the active zone membrane. We have added the pixel size for clarification. Indeed, we do not observe an increase in release probability or first evoked response, which would correspond with an increased docked pool. However, we think that the increase in docked vesicles might contribute to an enhanced SV recruitment (see discussion).

(5) Figure 3: Vesicle cycling was monitored in only a limited condition. It is known that there are multiple pathways of vesicle cycling. Ideally, these pathways should be dissected. At least, the authors mention the possibility that they have missed some "positive" conditions.

We fully agree with the reviewer’s comment that vesicle recycling is complex with several parallel pathways involved. While we did not study individual endocytosis pathways, we used different assays covering various recycling pathways. The SypHy assay (Fig. 3c & f) combined with the 100 AP stimulation paradigm at room temperature predominantly addresses clathrin-mediated endocytosis. Additionally, the FM-64 dye assay at 37 degrees Celsius covers ultrafast endocytosis pathways as well as bulk endocytosis routes. Since neither assay showed major effects, we decided not to pursue further experiments focusing on different endocytosis pathways.

**Reviewer #3 (Recommendations For The Authors):**
Major points:(1) Since all of the work here is culture-focussed, the in vivo phenotype is not as relevant, however the in vitro properties are. The incomplete Cre-dependent removal of SNX4 is concerning (especially axonal SNX4 levels identified via immunofluorescence), however, the main concern is that there was no profiling of the other molecular changes within these cultures. This is important, since there may be considerable alterations in the expression of a number of presynaptic proteins which may explain the observed phenotypes. Ideally, these cultures could have been profiled in an unbiased manner via mass spectrometry to identify potential changes in the presynaptic proteome, or at the very least the levels of key fusion molecules would have been assessed via Western blotting.

We thank the reviewer for their suggestion and agree that mass spectrometry would strengthen the interpretation of the observed phenotype. However, due to contractual constraints, we are unable to pursue a mass spectrometry follow-up experiment. We agree that characterizing key fusion molecules is of potential interest. Therefore, based on literature, we selected a likely candidate, VAMP2, which did not show any alterations in expression levels when knocking out SNX4. Given the previously described role of SNX4 in the degradation pathway, one would expect increased degradation of key fusion molecules if they are recycled by SNX4. Other literature indicates that reduced levels of key fusion molecules, such as synaptotagmin or SNAP-25 (Broadie et al., 1994; Washbourne et al., 2001) , do not mimic our phenotype.

(2) The experiments reported in Figure 2, in particular those in 2c and 2d, suggest that overexpression of SNX4 has a dominant-negative effect on neurotransmitter release. This is strongly supported by the supplementary data during a stimulus train (particularly the start point of the 5 Hz train in Supplementary Figure 2). Therefore, the perceived rescue of EPSC charge in Figure 2f, 2g may be a result of SNX4 inhibiting neurotransmitter release. A determination of the impact of SNX4 overexpression (and level of overexpression) in WT neurons is essential to show that this is a bonefide rescue, rather than a direct inhibition by SNX4 overexpression.

We thank the reviewer for their suggestion and agree that an additional experimental group (control + SNX4) would strengthen interpretation of the observed phenotype. We have now added a new experiment with an extra experimental condition with overexpression of SNX4 on a control background (Supplementary Fig. 3 page 21). This data shows that the amplitude and charge of the first evoked response were not affected in control + SNX4 neurons compared to control, and no differences were detected in the response to the 40 Hz stimulation train (Supplementary Fig. 3a-e). Together, these data suggest that SNX4 overexpression in itself does not affect the neurotransmission protocols studied in SNX4 cKO experiments.

(3) The experiments in Figure 3 clearly reveal a lack of effect of SNX4 depletion on synaptic vesicle endocytosis. However, the assumption that synaptic vesicle recycling is unaffected is a *little premature*. The fact that the second evoked SypHy peak is significantly larger than the first (Figures 3c-e) suggests that more vesicles may be recycling in KO neurons. Furthermore, the FM dye experiments do not aid interpretation, since there may be insufficient time (10 min) for new vesicles to be generated from endosomal intermediates experiments. Therefore, to confirm an absence of effect on recycling, the authors could either (1) perform the same experiment as 3c, but with 4 stimulation trains (to drive the system harder to reveal any phenotype) or (2) repeat the FM dye experiment but increase the time between loading and unloading to 30 min.

We fully agree with the reviewers' comment that vesicle recycling is an important component to consider and is complex with several parallel pathways involved. We conducted multiple independent experiments covering the most significant recycling pathways. The SypHy assay (Fig. 3c & f) combined with the 100 AP stimulation paradigm at room temperature predominantly addresses clathrin-mediated endocytosis. Additionally, the FM-64 dye assay at 37 degrees Celsius covers ultrafast endocytosis pathways as well as bulk endocytosis routes. To further challenge the system and reveal recycling phenotypes, we included a second 100 AP stimulation in our SypHy assay. While only the increase of the second SypHy peak is significant, the absolute numbers do not differ much from the first peak (0,17 for control and 0,21 for KO second peak and 0,19 for control and 0,22 for KO first peak, Supplementary table1). We nevertheless do not see any effects on recycling after the second peak (mean decay time is 27 for control and 26 for KO Supplementary Table 1). A single 100 AP 40 Hz train depletes all the synchronous release (not shown) and most of the evoked charge (see Fig 2f), hence two of these trains with one minute recovery is already a very demanding protocol. Although increasing the time between loading and unloading to 30 minutes might uncover other recycling components, it has been shown that ultrafast endocytosis occurs within 30 seconds (Watanabe et al., 2013), suggesting that 10 minutes should provide enough time for synaptic vesicle recycling. This is also evident from the fact that we can significantly destain synapses loaded with FM dye by electrical stimulation (Fig 3j), indicating that synaptic vesicle recycling took place. Since neither assay showed major effects, we concluded that under these circumstances, synaptic recycling is not significantly affected. However, we cannot exclude the possibility that recycling deficits in SNX4 cKO neurons could be detected in other paradigms,

(4) There is no obvious effect on VAMP2 levels or location in SNX4 KO neurons (Figure 4). However, when one considers that SNX4 is proposed to have a role in VAMP2 trafficking, it is surprising that an experiment examining the live trafficking of VAMP2-SypHy was not performed. This would have revealed activity-dependent alterations that would have been missed by simply measuring VAMP2 expression and localization, and potentially provided a molecular explanation for the enhanced neurotransmitter release during a stimulus train.

We appreciate the reviewer’s suggestion and agree that it could be a valuable experiment However, overexpressing a VAMP2-pHluorin construct might obscure potential phenotypes related to VAMP2 trafficking. SNX4 is expected to be involved in VAMP2 recycling, even with activity-dependent changes. Mis-sorted VAMP2 would accumulate in acidic vesicles, which could be masked by the VAMP2-pHluorin construct. Similarly, mis-sorting of other SNX4 cargo, such as the transferrin receptor, has been identified through lysosomal degradation, as shown by Western blot analysis of expression levels of the endogenous protein. We did not detect any differences in endogenous levels of VAMP2 within 21 days of SNX4 deletion (Fig 4), indicating that SNX4-dependent endosome sorting is not essential for VAMP2 recycling.

(5) The morphological data in Figure 5 report a series of small changes in docked vesicles and active zone length. In many cases, significance is obtained due to synapses being used as the experimental n, and thus inflating the statistical power. When one considers that no significant effect was observed on evoked release (apart from during a stimulus train), it suggests that the number of docked vesicles does not alter release probability in this system (which the authors point out). Instead, they suggest that an increased supply of vesicles is responsible, via increased recruitment to RRP/releasable pool (but not via increased recycling). If this is the case, it should have been reflected as an increase in the evoked SypHy response in Fig 2c,d (which is borderline significant). What may help is to determine the morphological landscape immediately after a stimulus strain, since this is the only condition where enhanced release is observed, and thus provide a morphological correlate to the physiological data.

We fully agree with the reviewer’s suggestion that an ultrastructural characterization immediately after a stimulus train would be informative. Unfortunately, contract constraints prevent us from performing this experiment. For our ultrastructural morphological data, we treated synapses as individual experimental *n* since it is not possible to determine whether synapses in a micronetwork on one sapphire originate from the same neuron. We used 18 independent sapphires from 3 independent pups to ensure the technical and biological replication of our data and measuring independent neurons. We fully agree with the reviewers comment to be careful with ‘inflating the statistical power’ due to potential nesting effects when using synapses as experimental n. To mitigate the potential nesting effect of analyzing multiple synapses per neuron, the intracluster correlation (ICC) is calculated per variable and per nesting effect. If ICC was close to 0.1, indicating that a considerable portion of the total variance can be attributed to e.g. synapse or sapphire, multilevel analysis was performed to accommodate nested data (Aarts et al., 2014).

Minor points(1) When a new mouse model is generated, it is usually accompanied by a thorough characterization of its properties. However, in this case, there was no information provided about the conditional SNX4 knockout mouse. This is surprising and at a minimum, the following should be provided (a) the background strain, (b) method of generation, (c) the number of animals used to establish the colony, (d) breeding strategy, (e) backcrossing strategy, (f) genotyping protocol.

We apologize that a thorough characterization of our novel mouse model was lacking and therefore added this to our material & methods section (page 11, line 377-391).

(2) There is a noticeable difference between WT and KO neurons during train stimulation in Figure 2f, however, this appears to be due to the fact that there is a far higher EPSC charge to begin with in KO neurons. Why is there such a disparity when there is no difference in response to single pulses (Figures 2b-d) or presynaptic plasticity (Figure 2e)?

We understand the reviewer’s concern. We excluded an outlier (3x SD) in the KO dataset that drove the initial far higher EPSC charge in the graph (was already excluded for the statistics, Supplementary table 1). The average charge of the first pulse of 40Hz train is 41 pC and for KO neurons 58 pC, which did not differ significantly. These trains of Fig. 2f were recorded after 2 or 3 other stimulation paradigms, which can have affected the total charge released in the 40Hz train. That said, the proportional difference between groups is high comparable between Fig 2b-d and 2f, with a 37% increased average charge released in SNX4 cKO compared to control in the naïve response (Fig. 2d) and 41% increased response in the first response of the 40 Hz train (Fig. 2f), and rescued cells show a 53% reduction in average released charge compared to control in the naïve response compared to a 44% reduction in the first response of the 40 Hz train. Although the absolute values differ between these readouts, we conclude that the biological comparison between groups is consistent.

(3) Line 343-344 - "(Supplementary Figure 1a)" should be "(Figure 1a)".

We thank the reviewer for this comment and adjusted this in the manuscript.